# Fast Fabrication of Fishnet Optical Metamaterial Based on Femtosecond Laser Induced Stress Break Technique

**DOI:** 10.3390/nano11030742

**Published:** 2021-03-16

**Authors:** Kai-Xin Zhang, Jian-Da Shao, Guo-Hang Hu, Ying-Jie Chai, Hong-Bo He, Mei-Ping Zhu, Da-Wei Li, Xiao-Feng Liu

**Affiliations:** 1Laboratory of Thin Film Optics, Shanghai Institute of Optics and Fine Mechanics, No. 390 Qinghe Road, Jiading District, Shanghai 201800, China; kxzhang@siom.ac.cn (K.-X.Z.); hbhe@siom.ac.cn (H.-B.H.); bree@siom.ac.cn (M.-P.Z.); lidawei@siom.ac.cn (D.-W.L.); liuxiaofeng@siom.ac.cn (X.-F.L.); 2Center of Materials Science and Optoelectronics Engineering, University of Chinese Academy of Sciences, Beijing 100049, China; 3Key Laboratory of Materials for High Power Laser, Chinese Academy of Sciences, Shanghai 201800, China; 4CAS Center for Excellence in Ultra-intense Laser Science, No. 390 Qinghe Road, Jiading District, Shanghai 201800, China; 5Hangzhou Institute for Advanced Study, University of Chinese Academy of Sciences, Hangzhou 310024, China; 6CREOL, The College of Optics and Photonics, University of Central Florida, Orlando, FL 32816, USA; yjchai@ucf.edu

**Keywords:** femtosecond laser, fishnet metamaterials, nanofabrication, stress break, negative reflective index

## Abstract

To speed up the fabrication of optical metamaterials by making use of the fast speed advantage of femtosecond laser preparation, a metamaterial appropriate for femtosecond laser processing was designed, and the interaction between femtosecond laser and metal-dielectric-metal fishnet stacks was investigated in detail. Two kinds of processing mechanisms, thermal melting and stress break, were revealed during the fabrication. The thermal melting process, dominated by the interaction of femtosecond laser with metals, makes the upper and lower metal layers adhere to each other, which leads to the magnetic resonance impossible. The stress break process, dominated by the interaction of femtosecond laser with dielectrics, can keep the upper and lower metal coatings isolated. Fishnet optical metamaterial was fabricated by femtosecond laser-induced stress break technique, using back side ablation, high numerical aperture and super-Gaussian beam. The resolution and speed can reach 500 nm, and 100 units/s, respectively. Spectrophotometer measurement results proved that the magnetic resonances were found in the fishnet nanostructure. The theoretical refractive index of the metamaterial on a glass substrate reached −0.12 at the wavelength of 3225 nm. It proved that femtosecond laser-induced stress break was a good and fast tool during the fabrication of optical metamaterials.

## 1. Introduction

Metamaterials (MMs) which have extraordinary properties have attracted much attention from researchers, like negative complex refractive index [1], extraordinary optical transmission [2,3], enhanced absorption [4], and super diffraction limit focusing [5]. MMs with a negative refractive index (NRI) can make a cloak [6,7] and super diffraction limit imaging [8] come true by systematically manipulating light. Since split ring resonators were designed [9], the structure of NRI has changed from upstanding split-ring resonators [10] and double-layer of cut-wires [11] into fishnet structure [12], periodic holes penetrating metal-dielectric-metal (MDM) stacks, for simple fabrication process and reduced loss. Fishnet MMs, a metal-dielectric-metal three-stack structure with transversely subwavelength holes, are the most likely and the first structure [12] to achieve NRI in the visible spectrum and near-infrared band. The practical application of these fishnet MMs in optical frequencies encountered the following challenges. Firstly, the substrate definitely affects the NRI of fishnet MMs and few reports consider this substrate influence. Secondly, it is still difficult to find a fast, feasible and nano-resolution fabrication method of fishnet optical MMs.

Traditionally, fishnet MMs are fabricated by electron beam lithography [13,14], focused ion beam lithography [15,16,17], nanoimprint lithography [18], and interference lithography [19]. Femtosecond (fs) laser direct writing technology, a maskless fabrication technique, provides a possibility of fabricating fishnet MMs with high speed and nanometer resolution. It takes a few hours to fabricate a structure of 100 µm × 100 µm, while electron beam lithography [13,14] needs long time. Until now, there are few reports about optical fishnet MMs fabricated by fs laser technology. This is mainly because the mechanism of fs interaction with metal-dielectric-metal stacks is not particularly clear.

Mechanisms of interaction between fs laser and dielectrics or metals are relatively clear. The main cause of the interaction process between fs laser and dielectrics is photoionization. Electrons absorb energy nonlinearly and are excited from the valence band to the conduction band. When the density of laser-induced electrons reaches the plasma critical density, a micro-explosion happens, and the ablated region breaks down. As the fs pulse duration is short and electrons in dielectrics cannot move freely, the electron-lattice interaction does not take place during the interaction process and the damage is limited to a small region [20]. The interaction process between fs laser and metals involves a thermal process. The free electrons absorb energy from the electric field and are heated to an extremely high temperature, which is above the vaporization temperature. After the pulse, laser-induced electrons transfer energy to the lattice by electron-lattice scattering, causing a local thermal damage [21]. However, there are few reports on the mechanism of fs laser interaction with the MDM films. How to suppress the thermal process is also a challenge in fs laser fabrication.

In this paper, a MM structure on a substrate, suitable to fs laser fabrication, was firstly designed. Two kinds of fs interaction mechanism with MDM coatings, thermal melting and stress break, were revealed. In particular, the laser-induced stress break process could inhibit the thermal diffusion and thermal damage of the fishnet structure, making the upper and lower metal stacks isolated. The laser-induced stress break made the magnetic resonance possible. By fully taking advantage of the stress break process, a subwavelength periodic hole array (105 μm × 105 μm) in MDM films with a period of 1.5 μm was fabricated by fs laser pulse with back side ablation, large numerical aperture, and super-Gaussian beam. The fabrication speed and resolution have reached 100 units/s and 500 nm, respectively. Transmittance and reflectance of this fishnet metamaterial were measured to prove that the fishnet fabricated by fs laser ablation generated magnetic resonances. A theoretical negative refractive index of −0.12 at the wavelength of 3225 nm was achieved in the fishnet metamaterial placed on a glass substrate, which made fs laser-induced stress break technique possible for optical metamaterial fabrication.

## 2. Materials and Methods

### 2.1. Materials

The Au single layer was deposited by magnetron sputtering technique on a glass substrate. The MgF_2_ single layer was deposited by electron beam evaporation on the glass substrate.

### 2.2. Morphology Characterization

Surface morphology of subwavelength nanoholes in Au films, MgF_2_ films, and the Au-MgF_2_-Au films were characterized by scanning electron microscopy (SEM, Auriga, Carl Zeiss AG, Jena, Germany).

### 2.3. Spectral Characterization

The spectral transmittance and reflectance of the fabricated fishnet metamaterials were measured in the near-infrared region by Fourier Transform Infrared Spectrometer (Nicolet iS50, Thermo Fisher Scientific Corp., Waltham, MA, USA). The plane wave was incident normally on the sample.

### 2.4. Theorectical Simulation of the Fishnet MM for fs Laser Fabrication

As shown in Figure 1, a model of a fishnet structure with circular holes, was designed to simulate the transport of light in the fishnet metamaterial. The model consisted of Au-MgF_2_-Au stacks with transversely circular holes. The thickness of Au layer and MgF_2_ layer were 45 nm and 30 nm, respectively. The input medium was air, and the output medium was a glass substrate with a refractive index being 1.57. The period of circular holes was 1.5 μm, and the diameter of the holes was about 930 nm. Among metals, Au and Ag have the lowest damping in the optical frequencies. However, Ag is oxidizable and Au is more stable for long-term application. Therefore, Au was used in this structure. The circular hole is appropriate for fs laser fabrication and polarization-independent (both for the *x* direction and *y* direction). A dielectric with a relatively small permittivity can obtain a more negative effective permeability, so that MgF_2_ with the refractive index of 1.38, was chosen as a spacer. A free electron Drude model: 
$\varepsilon (\omega )=\varepsilon_{\infty }-\frac{{\omega_{P}}^{2}}{\omega^{2}+i\Gamma \omega }$
was applied to describe the dispersion of the Au layer. *ε*_∞_ was the offset value of permittivity with the value of 1, *ω* was the frequency of the incident plane wave, *ω**_p_*= 10^16^ was the plasma frequency and *Γ* = 2.07 × 10^14^ s^−1^ was the collision frequency [11,22]. In the theoretical model, a polarized plane wave was incident normally on the fishnet in the spectral range of 1700 nm to 4000 nm. The optical response of the fishnet stack, transmittance *t* and reflectance *r* were obtained by the calculation of finite element analytical method.

The optical properties of fishnet were achieved by retrieval algorithm. The improved analytical method included the influence of the substrate, compared to D. R. Smith’s result [23,24]. The fishnet structure can be seen as a homogenous slab to retrieve effective optical constants from the transmittance *t* and reflectance *r* by an improved analytical method. For a TE mode wave (*s*-polarized light), the characteristic matrix of fishnet structure is,
(1)$$M=(\begin{array}{cc} m_{11} & m_{12}\\ m_{21} & m_{22} \end{array})=\left(\begin{array}{cc} cos(k_{0}nd\;cos\theta ) & -\frac{i}{p}sin(k_{0}nd\;cos\theta )\\ -ip\;sin(k_{0}nd\;cos\theta ) & cos(k_{0}nd\;cos\theta ) \end{array}\right)$$
where *k*_0_ is the wave vector in the air of the incident plane wave, *n* is the refractive index of the fishnet structure, *d* is the thickness of the fishnet structure, *θ* is the incident angle of the plane wave, and 
$p=\sqrt{\frac{\varepsilon }{\mu }}cos\theta$
, *ε* and *μ* 
are the permittivity and permeability of the fishnet. For a TM mode wave, $q=\sqrt{\frac{\mu }{\varepsilon }}cos\theta$ is substituted for *p*.

For TE mode, the reflectance and transmittance coefficients *r* and *t* are described as
(2)$$r=\frac{\left(m_{11}+m_{12}p_{b})p_{a}-(m_{21}+m_{22}p_{b}\right)}{\left(m_{11}+m_{12}p_{b})p_{a}+(m_{21}+m_{22}p_{b}\right)}$$
(3)$$t=\frac{2p_{a}}{\left(m_{11}+m_{12}p_{b})p_{a}+(m_{21}+m_{22}p_{b}\right)}$$
where
$p_{a}=\sqrt{\frac{\varepsilon_{a}}{\mu_{a}}}cos\theta_{a}$, 
$p_{b}=\sqrt{\frac{\varepsilon_{b}}{\mu_{b}}}cos\theta_{b}$, *ε_a_* and *ε_b_* are the permittivity of the input material and the output material, *μ_a_* and *μ_b_* are the permeability of the input material and the output material, *θ*_a_ is the incident angle when
the plane wave is incident from the input material into the fishnet structure, and *θ_b_* is the incident angle when the plane wave is incident from the material into the output material. For TM mode, *p_a_* and *p_b_* are replaced by $q_{a}=\sqrt{\frac{\mu_{a}}{\varepsilon_{a}}}cos\theta_{a}$, 
$q_{b}=\sqrt{\frac{\mu_{b}}{\varepsilon_{b}}}cos\theta_{b}$.

For a normally incident electromagnetic wave (*θ* = 0°), the refractive index is derived as:(4)$$cos(nk_{0}d)=\frac{t^{2}p_{b}+p_{a}(1-r^{2})}{p_{b}t(1+r)+p_{a}t(1-r)}.$$
(5)$$n=\frac{{cos}^{-1}(\frac{t^{2}p_{b}+p_{a}(1-r^{2})}{p_{b}t(1+r)+p_{a}t(1-r)})}{k_{0}d}$$


Simulation results of transmittance and reflectance of the above fishnet structure are shown in Figure 2. At the wavelength of around 2300 nm (the yellow arrow in Figure 2), there are an obvious valley of the reflectance curve and a peak of the transmittance curve. The reflectance curve shows two slight valleys at the wavelength of about 2100 nm (the red arrow in Figure 2) and 3220 nm (the green arrow in Figure 2). As shown in Figure 3, optical constants like the refractive index, permittivity and permeability were retrieved by the above improved analytical method from the reflectance ***r*** and transmittance *t*. NRI was achieved around the valley from 3076 nm to 3390 nm, with the minimum value of −0.12 at 3225 nm. This paves a way for the real application, such as super diffraction imaging in the mid-infrared band, of fishnet metamaterials placed on a substrate. Between 1700 nm and 2300 nm, there is also a valley in the curve of refractive index. However, the minimum value of the valley is larger than zero. The two valleys result from magnetic resonances (Figure 3b). The valleys at the longer wavelength along with a more negative permittivity generated NRI. At around 2300 nm, there is a peak in the curve of refractive index. This corresponds to the electric resonance (Figure 3c), which induces an anti-magnetic resonance. The anti-magnetic is not wanted because it induced an increased refractive index.

Figure 4 and Figure 5 show the distributions of the magnetic fields at the wavelength of magnetic resonance (3225 nm and 2072 nm) in the in the *x-z* plane and *x-y* plane in the MgF_2_ layer (*z* = 0). The *x-z* plane can be regarded as a *LC* circuit, and the metal and the dielectric layer can be seen as the inductance and the capacitance, respectively [25]. As shown in Figure 4, at the magnetic resonance frequency, the incident electromagnetic excites a loop current in the upper and lower interface between the metal layer and the dielectric layer, and produces the magnetic resonance. The electromagnetic is confined in the fishnet metamaterial. As shown in Figure 5, the magnetic fields at 3225 nm and 2072 nm are both focused in the fishnet structure. The magnetic fields at 3225 nm are more confined around the hole, while the magnetic fields at 2072 nm are enhanced at the intersections of the two perpendicular strips in the *x* and *y* direction. As shown in Figure 3, at the magnetic resonance of 3225 nm, the permeability and the permittivity are both smaller than those at 2072 nm, respectively, which results to the NRI.

### 2.5. fs Laser Fabrication of Fishnet MMs

The fs laser fabrication setup is shown in Figure 6. The setup includes three systems: fs laser fabrication system, on-line imaging system, and automatic controlling system. The laser was at the wavelength of 520 nm, and the duration was 350 fs. The repetition frequency of the laser was 100 kHz. In the fs laser fabrication system, the linearly polarized laser pulse firstly went through an attenuator consisting of a half-wave plate and a polarizer. Then the laser beam was expanded by 6 times and went through an aperture whose diameter was 500 μm. And then the beam was expanded for the second time by 20 times. The central part of the Gaussian beam passed the second aperture with a diameter of 6 mm to generate a super-Gaussian beam. Finally, it was focused on the sample by an objective lens (Olympus), whose numerical aperture (NA) was 1.4. Figure 7a shows the profile of the fs laser beam before passing the first expander of the experimental setup described in Figure 6. Figure 7b shows that fs laser beam profile after passing through the second aperture in the setup. The sample was placed on a 3-dimension moving stage, which had a resolution of 50 nm in the ***x*** and ***y*** directions. The white light penetrated through the sample, collimated by the high NA lens and then focused by a tube lens. The fabrication process could be observed online. In the fabrication process, the automatic controlling system could control the energy and the repetition frequency of laser pulse, and the movement of the 3-dimension moving stage and image.

Resolution is closely related to NA of a lens, because a lens with a large NA can focus the light beam tightly. NA could be large than 1, when the lens immerses in oil, which inevitably ruins the sample. To increase NA and prevent samples from polluting, a back side ablation technique was developed. As shown in Figure 8, back side ablation is a fabrication method that the light is incident from the glass substrate side and then is focused on the bottom surface of films. The substrate is immersed in oil and the film is exposed to the air. Therefore, the film stack and subwavelength structure will not be destroyed by the immersion oil.

To show mechanisms of interaction of fs laser with Au-MgF_2_-Au films, Au single layer coating and MgF_2_ single layer coating were also ablated by fs laser with a period of 1.5 μm. Therefore, the samples were a 45-nm Au film, a 30-nm MgF_2_ film, and a 120-nm Au-MgF_2_-Au film with the size of 18 mm × 18 mm. The Au-MgF_2_-Au film stacks were consisting of an upper and a lower layer of 45 nm Au films and a layer of 30 nm MgF_2_ film, respectively, the same as the designed MDM film structures. Their roughness is all below 1 nm. As shown in Figure 9, Au films were sputtered on a glass substrate by magnetron sputtering technique and MgF_2_ films were deposited by electron beam evaporation. The glass substrate had the size of 18 mm × 18 mm × 0.16 mm, and the refractive index is 1.57, the same as the designed structure. Fishnet MMs were fabricated on the MDM film stacks by fs laser beam according to the designed structure with energy of 7.8 nJ and 26.4 nJ. The structure fabricated on single Au layer and single MgF_2_ layer were ablated by fs laser beam with the energy of 2.17 nJ, and 29.33 nJ, respectively.

## 3. Results and Discussions

Periodic nanoholes in Au single layer, MgF_2_ single layer, and Au-MgF_2_-Au film stacks were ablated by fs laser and the geometry were characterized by scanning electron microscopy.

Transmittance and reflectance measurements of the fishnet metamaterial, with a periodic square array of circular holes fabricated by stress break technique with a fs laser, were made.

### 3.1. Geometry of Au Film, MgF2 Films, and Au-MgF2-Au Film Ablated by fs Laser

Since fishnet structure with circular holes shows a better performance, suitable for fs laser fabrication and polarization-independent both in the *x* and *y* direction, periodic round nanoholes in Au-MgF_2_-Au film stacks were ablated by fs laser with different energy, which is shown in Figure 10. The morphology induced by the fs laser beam with a smaller pulse energy of 7.8 nJ is shown in Figure 10a. There is a large area of fused rough ripples around the nanoholes. It is more like the morphology induced by fs-metal interaction. From the image of cross section (Figure 10b), the melting edges of the upper Au layer and the lower Au layer greatly protrude upwards, therefore the upper and the lower metal films adhere to each other, which would inhibit the magnetic resonant effect of the fishnet structure. Figure 10c shows the morphology induced by the fs laser beam with a higher pulse energy of 26.4 nJ. The diameter of each hole is around 700 nm. Obviously, the edges of the periodic holes are relatively smooth, with little fused region. Each hole seems to be created by stress break, leaving a hollow bulge. The hole was produced by two steps. In the first step, a bump was created by the shock of laser beam. Then, the bump became higher and higher, until it broke down by a hard stress and the center of the bump disappeared. Therefore, the edge of the hole was smooth with small roughness. Moreover, the image of cross section (Figure 10d) suggests that the upper and lower metal films are not jointed together, making magnetic resonant phenomenon possible. To clearly illustrate the mechanisms of fs laser interaction with MDM film stacks, periodic nanoholes in Au single layer and MgF_2_ single layer were also fabricated by fs laser and mechanisms of fs laser pulse interaction with metals and dielectrics were investigated.

Figure 11a,b shows the fs laser ablated geometry of periodic nanoholes in Au layer, and MgF_2_ layer, respectively. As shown in Figure 11a, there are large areas of fused ring around the nanoholes in Au layer, forming an unclear boundary between the holes and the surface. It mainly results from thermal melting rather than stress break. The minimal diameter of holes can be 500 nm. Two temperature model can explain the formation of holes with fused rings. For a short laser pulse, the pulse duration can be less than the relaxation time, that is *τ* < *τ_e__p_*, where *τ* is the laser duration, and *τ_ep_* is the relaxation time (the electron-phonon scattering time). During the pulse duration, the temperature of free electrons and the lattice are different and independent [26,27]. Free electrons absorb energy from the electric field. After the relaxation time, electrons transfer energies to the lattice by electron-phonon scattering. Most of the laser energy is taken away by the metal vapor, leaving a little energy in the lattice. The left energy brings out thermal diffusion, causing the formation of melting edges.

Unlike the geometry of nanoholes in Au single layer, there are smooth and vertical edges around the holes in MgF_2_ single layer, leaving a distinct boundary between the holes and the surface, as shown in Figure 11b. The main cause of damage of dielectric film by fs laser is ionization process, which can realize a resolution of 100 nm. The electrons in the valence band nonlinearly absorb energy from fs pulse and jump to the conduction band, which is called multiphoton ionization [28]. In this case, MgF_2_ has a big bandgap with 6.2 eV [29], and the energy of each photon (λ = 520 nm) is 2.4 eV. Therefore, an electron in the valence band of MgF_2_ absorb 3 photons to be excited to the conduction band. When the density of electrons is more than the plasma critical density, the plasma explosion happens, ejecting ions and leaving local damage in the radiation region [30]. This nonlinear process involves little thermal process, explaining the smooth edge of the hole in MgF_2_ layer.

Based on the above analysis, there are two mechanisms of fs laser interaction with the MDM film stacks, thermal melting, and stress break, respectively. When the interaction pulse energy is smaller, the dominating mechanism is the thermal melting process. The fs interaction with the metal layer is dominating the thermal melting process. The melting of the two metal stacks results in adherence to each other, which can be seen by comparing the geometry of fs interaction with metals and that of fs interaction with MDM coatings. The dielectric almost does not play a role in fs interaction with MDM film stacks. On the other hand, when the pulse energy is higher, the stress break plays a major role. In the process of stress break, the electrons in the dielectric layer absorb femtosecond laser energy. When the density of electrons is more than the plasma density, the micro-explosion in the dielectric layer produces a shockwave in the time scale of picosecond [31]. The shockwave generates a strong outward pulling force, leading to the picosecond transient damage of the metal layer, and leaving a relatively smooth edge around the hole. The bump around the hole in the MDM stack structure, as shown in Figure 10d, proved the existence of outward pulling force.

### 3.2. Spectrum Measurement of Fishnet Metamaterial Fabricated by Stress Break

As shown in Figure 12, a fishnet metamaterial with a periodic square array of circular holes, the size of which was 105 μm × 105 μm, was fabricated by stress break technique with a fs laser. The fabrication speed can reach 100 unit/s. The period was 1.5 μm, and the diameter of the holes was about 930 nm, the same as the structure in the simulation. The circular holes had clear and round edges with a few whiskers, and they were almost the same size. The measured and calculated transmittance *t* and reflectance *r* of the fishnet metamaterial are shown in Figure 13. The spectral transmittance and reflectance of the fabricated fishnet metamaterials were measured in the near-infrared region by Fourier Transform Infrared Spectrometer. The plane electromagnetic wave was incident normally on the sample. The measured curve of the reflectance have two valleys at around 2100 nm and 3200 nm, which are the magnetic resonant regions. With regard to the transmittance, there are three valleys at around 2100 nm, 2400 nm and 3200 nm, respectively. The three valleys correspond to the two magnetic resonances and one electric resonance, which are in the same position to the calculated values. Therefore, the calculated and measured optical properties are in relatively good agreement at the magnetic resonance position, but there is some discrepancy in the amplitudes at the electric resonance position.

The values of the measured curve are smaller than those of the calculated curve at the electric resonance position. The measured electric resonance was weakened at the electric resonance position. It is because that the electric resonance is sensitive to the surface geometry of metal surfaces. This discrepancy comes from the whiskers of the holes, which affects the electric resonance. The free electrons oscillate on the metal surface, and this action is called surface plasma oscillations [32]. The oscillation frequency is $\omega_{p}=\sqrt{4\pi ne^{2}/m}$, where *n* is the electron density, *m* is the mass of an electron, and *e* is the charge of an electron. When the frequency of incident light matches with the oscillation frequency, the electric resonance happens, and the electrons oscillate in the light frequency. The electric field of the surface decays exponentially in the direction perpendicular to the metal surface, and the electrons are locally bounded on the surface (usually in the skin depth). Therefore, the electric field is extremely sensitive to the surface of the metal. That is the reason why whiskers of the holes strongly affect the electric resonance.

At the magnetic resonance position, the calculated and measured values coincide with each other. This is because the magnetic resonance, which is vital to the NRI, is relatively insensitive to the surface geometry of the metal. The magnetic unit is a strip consisting of the upper metal layer, the dielectric layer, and the lower metal layer. The magnetic unit is regarded as a circuit [25], with the metal layer seen as inductance and the dielectric layer seen as capacitance. The magnetic resonance is only related to the inductance and capacitance, which results from the intrinsic property of the films. As shown in Figure 4, the loop constants are in the upper and lower interface between the metal layer and dielectric layer, with not being affected by the imperfect structure of the hole. Therefore, the magnetic resonance, resulting to NRI, is almost unaffected.

The calculated and the measured spectrum results of the fishnet metamaterials match well. That suggests that a NRI of −0.12 of the fishnet MM on a glass substrate at the wavelength 3225 nm was indeed achieved. The fabrication of fishnet structures on a substrate with NRI by fs laser fabrication are greatly possible.

## 4. Conclusions

In this paper, a fishnet metamaterial on a glass substrate with a negative refractive index was rapidly fabricated by stress break technique using a fs laser. The fabrication speed and resolution can reach 100 units/s, and 500 nm, respectively. The theoretical refractive index of the metamaterial on a glass substrate reached −0.12 at the wavelength of 3225 nm. Two kinds of processing mechanisms, thermal melting and stress break, were revealed in the fabrication process. The thermal melting process is dominated by the interaction of fs laser beam with metals, while stress break process is dominated by the interaction of fs laser beam with dielectrics, which can keep the upper and lower metal coating isolated. The stress break technique provides a way to fabricate fishnet metamaterial. For the application of fishnet metamaterial in visible wavelengths, the laser fabrication system needs to be improved to upgrade the resolution and achieve a perfect structure of the hole.

## Figures and Tables

**Figure 1 nanomaterials-11-00742-f001:**
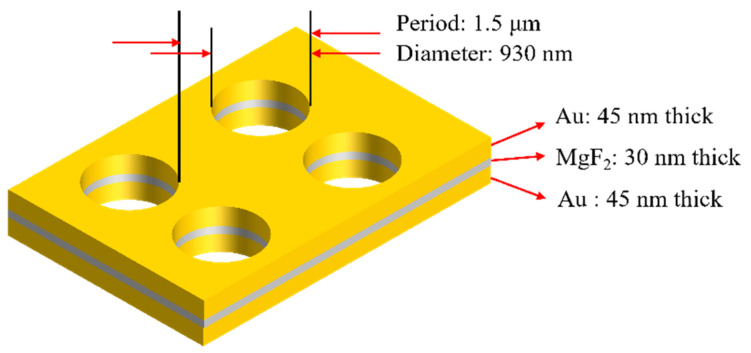
The diagram of theoretical model.

**Figure 2 nanomaterials-11-00742-f002:**
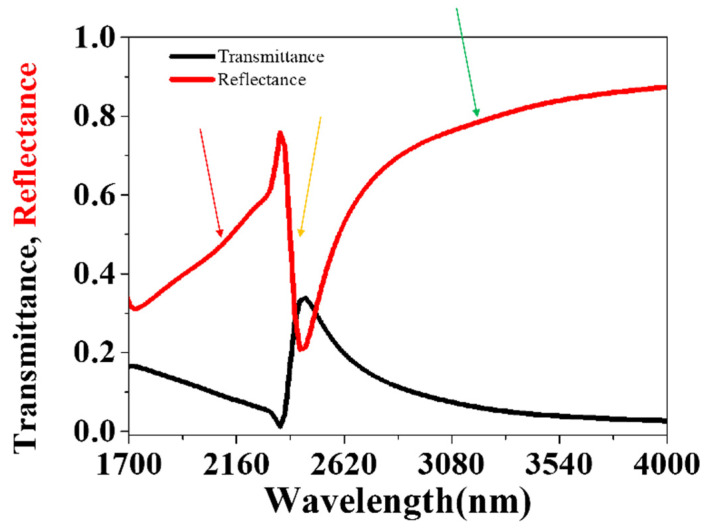
The simulated transmittance and reflectance curves of the fishnet metamaterial with circular holes. The red arrow points out the location of magnetic resonance at 2072 nm, the yellow arrow points out the location of electric resonance at 2300 nm, the green arrow points out the location of magnetic resonance at 3225 nm.

**Figure 3 nanomaterials-11-00742-f003:**
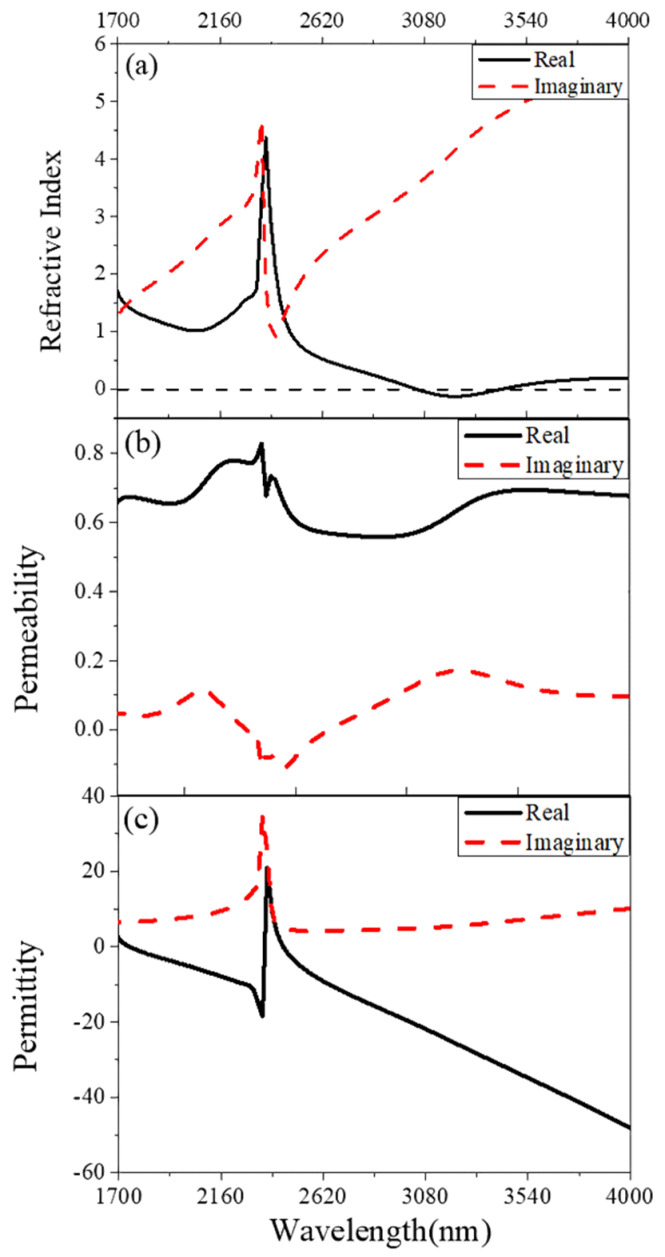
The simulated refractive index (**a**), the permeability (**b**) and the permittivity (**c**) of the fishnet MMs.

**Figure 4 nanomaterials-11-00742-f004:**
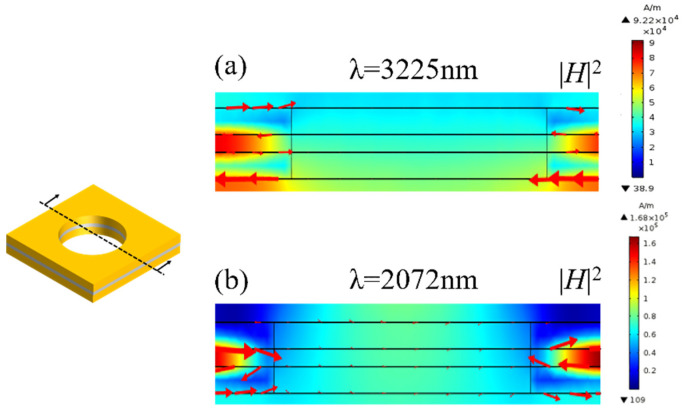
The magnetic field of the fishnet structure in the *x-z* plane in the resonant region of λ = 3225 nm (**a**) and λ = 2072 nm (**b**) (the arrow shows the direction of the current).

**Figure 5 nanomaterials-11-00742-f005:**
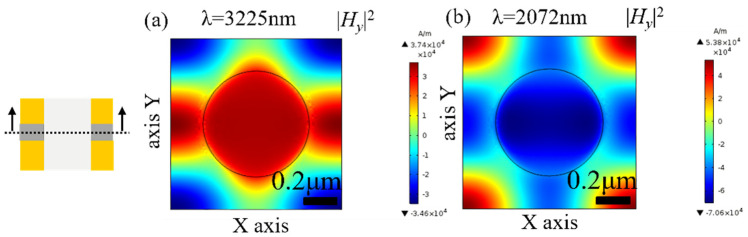
(**a**) The magnetic field distribution at λ = 3225 nm, (**b**) the magnetic field distribution at λ = 2072 nm in the *x-y* plane of the middle of MgF_2_ layer (*z* = 0).

**Figure 6 nanomaterials-11-00742-f006:**
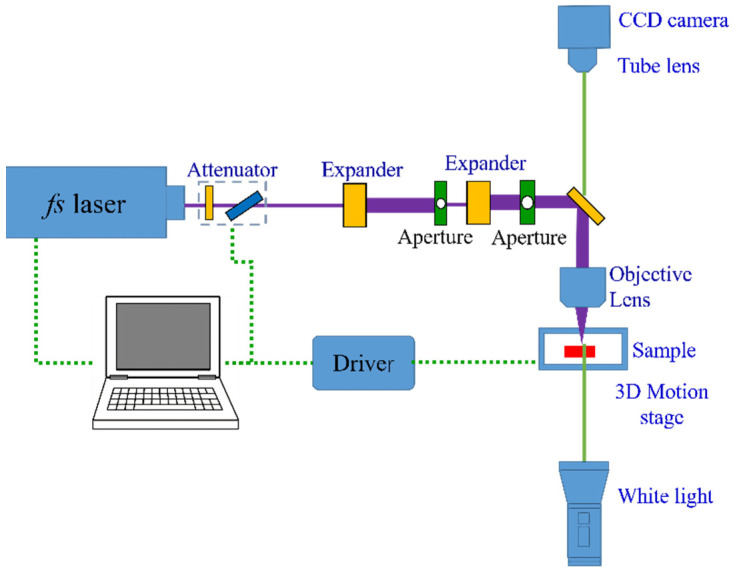
The schematic of femtosecond laser fabrication setup.

**Figure 7 nanomaterials-11-00742-f007:**
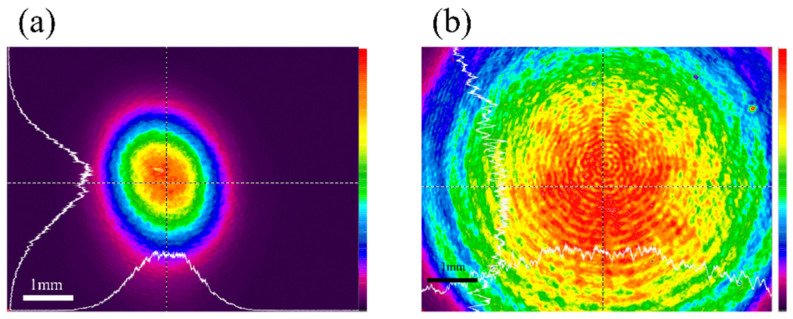
(**a**)The profile of the femtosecond laser beam before passing an aperture diaphragm; (**b**) the profile of the femtosecond laser beam after passing an aperture diaphragm.

**Figure 8 nanomaterials-11-00742-f008:**
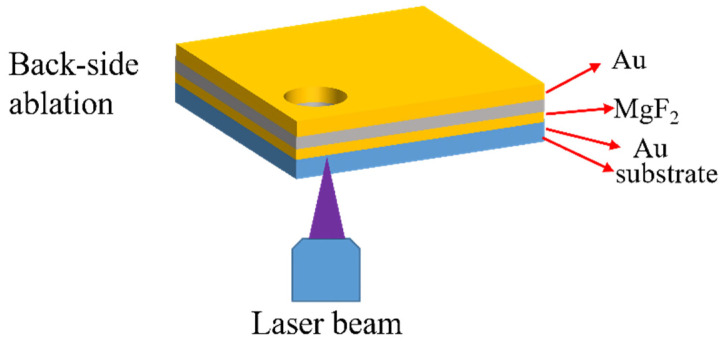
The diagram of back side ablation technology.

**Figure 9 nanomaterials-11-00742-f009:**
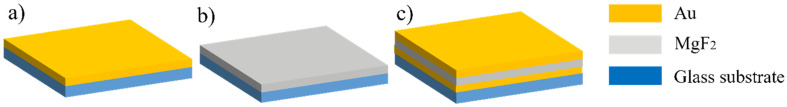
The diagram of Samples: (**a**) Au single layer deposited on a glass substrate; (**b**) MgF_2_ single layer deposited on a glass substrate; (**c**) Au-MgF_2_-Au film stacks deposited on a glass substrate.

**Figure 10 nanomaterials-11-00742-f010:**
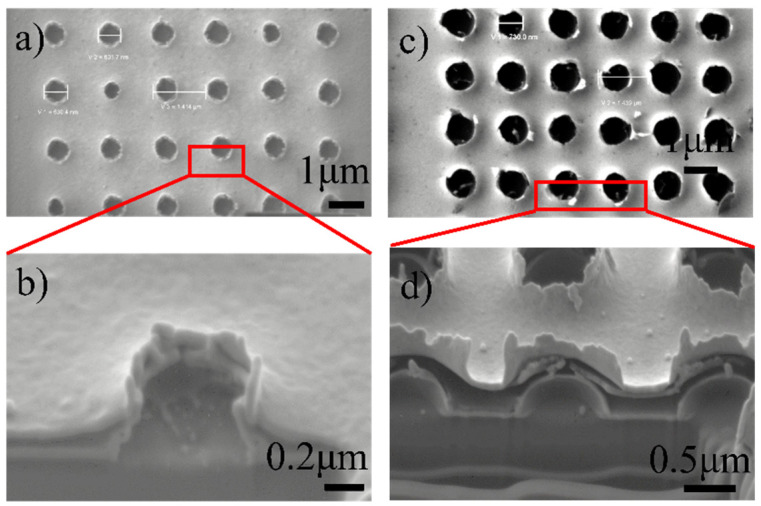
SEM images of periodic nanoholes in Au-MgF_2_-Au films ablated by fs laser: (**a**) fabricated by the fs laser beam with a smaller pulse energy of 7.8 nJ; (**b**) the cross section of the nanoholes of (**a**). (**c**) fabricated by the fs laser beam with a larger pulse energy of 26.4 nJ; (**d**) the cross section of nanoholes of (**c**).

**Figure 11 nanomaterials-11-00742-f011:**
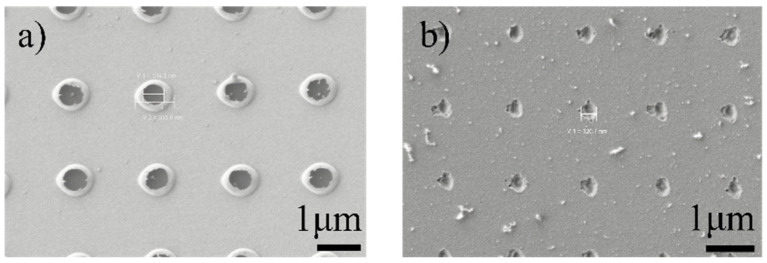
SEM images of periodic nanoholes in (**a**) Au film and (**b**) MgF_2_ film ablated by femtoseconds laser with different pulse energy.

**Figure 12 nanomaterials-11-00742-f012:**
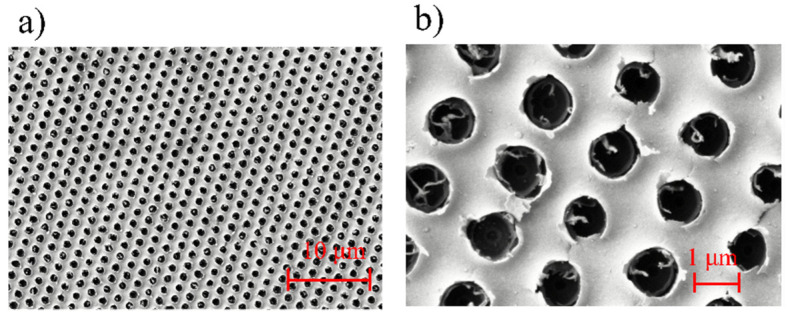
SEM images of the fishnet metamaterial with a periodic square array of circular holes with (**a**) the magnification was 2.5 k; (**b**) the magnification was 15 k.

**Figure 13 nanomaterials-11-00742-f013:**
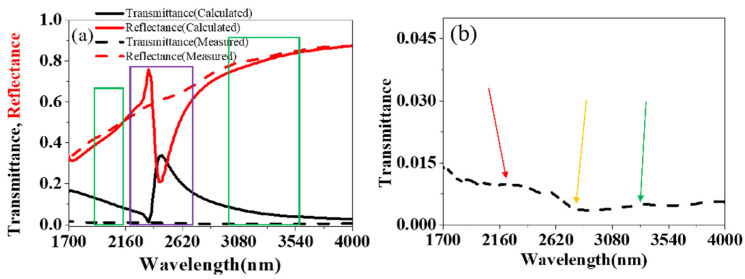
(**a**) The calculated and measured of transmittance and reflectance of the fishnet MDM, the green frames represent the position of magnetic resonances and the purple frame represents the position of electric resonance. (**b**) The magnified curve of the measured transmittance, the red arrow points out the location of magnetic resonance at 2072 nm, the yellow arrow points out the location of electric resonance at 2300 nm, and the green arrow points out the location of magnetic resonance at 3225 nm.

## Data Availability

Not applicable.

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
