# Peer review of "Fast Fabrication of Fishnet Optical Metamaterial Based on Femtosecond Laser Induced Stress Break Technique"

_nanomaterials, 2021, doi:10.3390/nano11030742_

Round 1
Reviewer 1 Report
See attached

Reviewer 2 Report
The paper is incomplete and somewhat disorganized. For example, Section 2.1 is 2 sentences with the 2nd sentence being "The text continues here." The modeling section goes into much more detail that needed; the magnetic field and electric field distributions are shown but with confusing descriptions and no further use of the information. There appears to be no attempt to do model optimization.
The experimental transmission and reflection results are quite disappointing. They only show a hint of the resonances predicted and far from the predicted magnitudes. The claim that a negative refractive index has been experimentally demonstrated is highly debatable.
The discussion of how the holes were drilled and the mechanisms is well done although the hypothesized mechanisms are often lacking experimental verification.
Further details on comments, questions and suggested grammar corrections are given in the attached pdf file.

Reviewer 3 Report
Suggest revising your caption in Figure 9 to talk about the parts in order of appearance or separate description of parts a) and c) from the description of parts b) and d) with a period. I also suggest doing a fine spell check for English and grammar
Round 2
Reviewer 1 Report
I would like to thank the authors for taking considerable time to revise their manuscript. This is much improved and highlights a fascinating study. This manuscript should be published without the need for any further revision.
Author Response
Dear Reviewer, We thank you very much for your supervision of the reviewing process and the opportunity to resubmit our manuscript, reference number nanomaterials-1114329 titled “Fast fabrication of fishnet optical metamaterial based on femtosecond laser induced stress break technique”. We highly appreciate your carefulness and conscientious comments, and the broad knowledge on the research field. We thank you once again for your work concerning our paper. Wish you all the best! Sincerely yours, Kaixin Zhang, Guohang Hu
Reviewer 2 Report
The authors have provided appropriate answers to my questions and comments in the response letter. However, some of their answers have not been incorporated into the manuscript. In particular figs 7 and 8 should be added with their discussions. The 'transient' discussion regarding point 9 should also be included in the manuscript.
